# Intergenic SNPs in Obstructive Sleep Apnea Syndrome: Revealing Metabolic, Oxidative Stress and Immune-Related Pathways

**DOI:** 10.3390/diagnostics11101753

**Published:** 2021-09-24

**Authors:** Dimitrios G. Raptis, George D. Vavougios, Dimitra I. Siachpazidou, Chaido Pastaka, Georgia Xiromerisiou, Konstantinos I. Gourgoulianis, Foteini Malli

**Affiliations:** 1Respiratory Medicine Department, School of Medicine, University of Thessaly, 41334 Larissa, Greece; raptisdmed@gmail.com (D.G.R.); gvavougyios@uth.gr (G.D.V.); sidimi@windowslive.com (D.I.S.); cpastaka@gmail.com (C.P.); kgourg@med.uth.gr (K.I.G.); 2Department of Neurology, School of Medicine, University of Thessaly, 41334 Larissa, Greece; georgiaxiromerisiou@gmail.com; 3Respiratory Disorders Lab, Faculty of Nursing, University of Thessaly, 41334 Larissa, Greece

**Keywords:** obstructive sleep apnea syndrome, genetics, metabolic, oxidative stress, immune-related pathways

## Abstract

There is strong evidence supporting the contribution of genetic factors to obstructive sleep apnea syndrome (OSAHS) susceptibility. In the current study we analyzed both in a clinical cohort and in silico, four single nucleotide polymorphisms SNPs, rs999944, rs75108997, rs35329661 and rs116133558 that have been associated with OSAHS. In 102 patients with OSAHS and 50 healthy volunteers, genetic testing of the above polymorphisms was performed. Polymorphism rs116133558 was invariant in our study population, whereas polymorphism rs35329661 was more than 95% invariant. Polymorphism rs999944 displayed significant (>5%) variance in our study population and was used in the binary logistic regression model. In silico analyses of the mechanism by which these three SNPs may affect the pathophysiology of OSAHS revealed a transcriptomic network of 274 genes. This network was involved in multiple cancer-associated gene signatures, as well as the adipogenesis pathway. This study, uncover a regulatory network in OSAHS using transcriptional targets of intergenic SNPs, and map their contributions in the pathophysiology of the syndrome on the interplay between adipocytokine signaling and cancer-related transcriptional dysregulation.

## 1. Introduction

Obstructive sleep apnea hypopnea syndrome (OSAHS) is a common disorder presenting with recurrent episodes of partial or complete upper airway obstruction that result in sleep disruption and intermittent hypoxemia with variable severity. On the genomic level, this pathophysiological substrate is linked with chronic upregulation of oxidative stress and hypoxia-responsive pathways and genes [1].

Individuals with OSAHS are at increased risk for cardiovascular disease, diabetes, stroke, cognitive impairment and many other disorders [2,3], and enhance morbidity and mortality rates [4].

The etiology of OSAHS is multifactorial, including the interplay between obesity, craniofacial structure, upper airway neuronal control, ventilator control and inflammation [5,6,7,8].

There is strong evidence supporting the contribution of genetic factors to OSAHS susceptibility [9]. Recent genome-wide association studies have revealed many hypoxia-signaling and sleep pathways [1]. Genome-wide association studies (GWAS) in OSAHS have been employed in order to discover genomic regions containing causal variants or variants influencing the expression of genes outside the identified region (e.g., enhancers and/or expression quantitative trait loci) [8]. The pathophysiological role of intergenic regions and pseudogenes is far less elucidated in several diseases, including OSAHS [10]. Specifically, no study to date has focused on the role of intergenic SNPs in its pathophysiology.

In the current study we aimed to elucidate the role of four intergenic SNPs selected from Cade et al.’s study [8]: rs999944 (genome-level significant association with AHI), rs75108997 (associated with sleep SpO_2_), rs35329661 (associated with event duration in females) and rs116133558 (associated with sleep SpO_2_), previously identified by Cade et al. [8] in a clinical cohort and via a standalone in silico workflow

## 2. Materials and Methods

This study was approved by the Institutional Review Board of University Hospital of Larissa (No: 63569-24 December 2018) and written informed consent was obtained from all participants.

### 2.1. Study Population

Our study included consecutive patients with suspected OSAHS who underwent polysomnography (PSG) test in the Clinic of Sleep Disorders of the Respiratory Medicine Department of the University General Hospital of Larissa were invited to participate in this study. All subjects underwent an overnight laboratory-based PSG, and the apnea–hypopnea index (AHI) was measured in all of them. OSAHS was defined as an AHI > 5 events/hr, and daytime symptoms specific for OSAHS. Hypopnea was defined as either (a) a >50% reduction in airflow, (b) a <50% reduction in airflow associated with a desaturation of >3% or (c) a moderate reduction in airflow accompanied by an electroencephalogram (EEG) -defined arousal. Patients were grouped according to the following classification by American Academy of Sleep Medicine (AASM 2007): mild disorder (AHI: 5–15 events/h), moderate disorder (AHI: 15–30 events/h), and severe disorder (AHI > 30 events/h). AHI < 5 events/hr was diagnosed as healthy subjects. A total of 102 Greek patients with OSAHS and 50 non-OSAHS, age- and gender-matched Greek controls were included in this study.

### 2.2. DNA Extraction and Genotyping

Genomic DNA was extracted from whole blood by DNA isolation kit, according to the manufacturer’s protocol. The purity and concentration of DNA were measured by a nanodrop spectrophotometer (Thermo Scientific, Waltham, MA, USA), with absorbance ratios ranging from 1.8 to 2.0 at the length of A260/A280. Genotyping of SNPs rs999944, rs75108997, rs35329661 and rs116133558 was performed using TaqMan single nucleotide polymorphism (SNP) genotyping technique on an ABI PRISM^®^ 7900 HT Fast Real–Time PCR System (Applied Biosystems, Waltham, MA, USA).

### 2.3. Statistical Analysis

All data analyses were performed with SPSS 23.0 (IBM Corporation, Armonk, NY, USA). Statistical significance was accepted at a level of *p* < 0.05. Deviation from the Hardy–Weinberg equilibrium was assessed using a chi-squared test with one degree of freedom. Statistical significance for categorical variables was assessed by the chi-squared or Fisher’s exact test. OSAHS risk and severity associated with the candidate SNPs were estimated by computing the odds ratios (ORs) and their 95% confidence intervals (CIs) by logistic regression analysis, adjusting for age, gender, and BMI. The analyses were done first per allele (allelic model) and then per genotype (additive model). Post-hoc power analysis was performed via the Bioinformatics Institute’s Online Sample Size Estimator (OSSE) (Available from: http://osse.bii.a-star.edu.sg/calculation2.php, accessed 16 September 2021) [11].

### 2.4. BLAST Analyses of Pseudogene and SNP Alignment in Intergenic Regions

SNPs mapped to pseudogenes and the corresponding pseudogenes themselves were analyzed for their potential biological activity in a multistep procedure. First, we determined that said pseudogene produced transcripts by inquiring the NHGRI-EBI GWAS catalog, a database of curated genome wide associated studies maintained by the National Human Genome Research Institute (NHGRI; Available from: https://www.ebi.ac.uk/gwas, accessed 16 September 2021) [12]. This resource allows the integration of both SNP and trait characteristics, as well as the integration of SNP data from multiple studies with other resources. As a next step, the Basic Local Alignment Sequence Tool (BLAST) [13] was used to detect overlap between a ±100 kb region [14,15] containing the intergenic SNP and pseudogenes within the same region. Subsequently, the Gene Cards database (Available from: https://www.genecards.org/, accessed 16 September 2021) [16] was mined for potential gene targets and transcription factor binding sites for each detected pseudogene.

### 2.5. Determination of SNP Interactions, and the Associated Genes’ Biological Networks

The SNPSnap tool (available from: https://data.broadinstitute.org/mpg/snpsnap/, accessed 16 September 2021) was used in order to identify interactions between SNPs, and identify common biological functions affected downstream [17]. 

Based on the above analyses and the results from the GeneCards database, we constructed a network of putative interactors from (a) SNP associated transcription factors (b) downstream genes (c) ARBB1, the gene mapped to the rs35329661 SNP (3′ UTR variant). This interactome that included a total of 274 genes was dubbed I_A_. Following the extraction of the I_A_ interactome, we performed gene set enrichment analyses via Enrichr web service (available from https://maayanlab.cloud/Enrichr/, accessed 16 September 2021), in order to predict its biological functions [18].

## 3. Results

### 3.1. Demographics, Clinical, Biochemical and PSG Characteristics of the Cohort

The main demographic and clinical characteristics of the 102 OSAHS cases and 50 healthy controls that underwent polysomnography are presented in Table 1 and Table 2. There were no significant differences between the OSAHS cases and control groups with respect to age or gender (*p* > 0.05).

### 3.2. SNP Variability and Multivariate Analyses of SNP-OSAHS Associations in the Clinical Cohort and Power Calculations

Polymorphism rs116133558, rs75108997 were invariant in our study population, whereas polymorphism rs35329661 was more than 95% invariant (C/C: 95.4%; C/T: 1.3; T/C:3.3%). Polymorphism rs999944 displayed significant (>5%) variance in our study population A/G: 20%; G/G:20%) and was used in the binary logistic regression model.

A binary logistic regression model adjusting for age, sex, BMI and lipid profile did not detect statistically significant associations between genotype frequency for the rs999944 SNP between OSAHS groups and controls (Table 3). Post-hoc analyses via OSSE for each SNP revealed that our study was under-powered (power < 50%).

### 3.3. In Silico Analyses of Gene and Pathway Associations of Non-Invariant SNPs

#### 3.3.1. BLAST Analyses

A BLAST analysis of the ±100 kb region containing rs116133558 revealed a hit in pseudogene AC114402.

Correspondingly, BLAST analysis of rs75108997 revealed total query cover and sequence similarity with pseudogene AL663058.5. The subsequent GeneCards Query did not reveal any significant transcription factor binding sites, or corresponding gene targets. The rs999944 displayed total query cover and sequence similarity with pseudogene AC007880.2. (Appendix A).

#### 3.3.2. Identification of SNP Gene Targets and Transcription Factors via GeneCards and SNPSEA

Interrogation of the GeneCards database revealed several gene targets and transcription factor bindings sites (TFBS) for rs11613358 and rs999944 (Table 4). Collectively, transcription factors (as genes) and gene targets were used to create a 274-gene signature that comprised interactome A (I_A_), the putative functional network of these two SNPS. SNPSEA did not reveal direct interaction between the two SNPS (Appendix A)

Subsequently, the I_A_ network (Appendix A) was used for GSEA in order to identify significantly enriched biological networks (Appendix A).

Table 5 and Table 6 report on the 10 first (sorted by adj.*p*-value) pathways resulting from GSEA on I_A_ via Enrichr (see Appendix A for the raw data).

## 4. Discussion

In this study, we explored the potential role of four intergenic SNPs in the pathogenesis of sleep apnea, both in a clinical cohort and in silico. Among the four selected variants, only rs999944 displayed significant variance in our population. No significant associations were detected between the genotype frequency of rs999944 and OSAHS biological parameters. In silico analyses of the mechanism by which these three SNPs may affect the pathophysiology of OSAHS revealed a transcriptomic network of 274 genes. This network was involved in multiple cancer-associated gene signatures, as well as the adipogenesis pathway.

### 4.1. Adipocytokine Signaling in OSAHS

The exposure of adipose tissue to intermittent hypoxia (IH) is an established perturbator in the pathophysiology of OSAHS [19]. In animal models of IH exposure, IH has been shown to prime adipocytes via C/EBP, disrupting PPAR and adiponectin signalling via adipocytokine release [20]. The sole study of visceral fat transcriptomes from OSAHS patients has corroborated this model in humans. More specifically, the PPAR system and adiponectin signalling cascades were disrupted in the setting increased proinflammatory signalling [21].

Our findings corroborate the findings of these studies on both the gene- and pathway-level. On the pathway level, the TNF, adipogenesis and adipocytokine signatures that we detected may reflect disruption of adiponectin signalling that has been correlated to adipocyte volume [19]. On the gene level, key targets of the rs116133558 and rs999944-associated pseudogenes have been previously reported as differentially expressed genes in IH models [19] and OSAHS [21], including PPARG, CEPA, CEPB, STAT3, RXRA and RXRB. Interestingly, the genotype frequency of rs999944 was independently associated with glucose levels in our clinical cohort, a relationship that could reflect decreased adiponectin signalling.

These findings indicate that the mechanism by which both intergenic polymorphisms contribute to OSAHS lies in their interaction with transcription factors and second-order phenomena, such as the disruption of regulatory networks [20], in contrast with other polymorphisms such as missense variants, where the disease association can be approached as a knock-out vs. wild type comparison.

### 4.2. Cancer-Related Networks in OSAHS

The epidemiological link between OSAHS and cancer is a complex one, potentially riddled with both clinical and biochemical confounders [22]. The main issue with the OSAHS—cancer connection is that both diseases are multifactorial and share both mechanisms [23] and comorbidities [24]. On the genomic level, this interplay may be further refined into sex- and site- specific correlates between OSAHS and cancer [25]. One of the few studies assessing the genetic components of OSAHS determined that cancer related pathways were affected by CPAP therapy [26].

Our findings corroborate with the latter study on both the gene- and pathway- level. Specifically, significantly enriched signatures that overlap between our studies included androgen signaling, breast cancer, and leukemia related pathways (see [20] and Appendix A; FDR < 0.05); Furthermore, key regulators of neoplastic processes and transcriptional dysregulation included genes such as JUN, SMAD3, MYC, HDAC1 and BRCA1 that were also part of the regulatory networks associated with rs999944.

Within this context, our findings support a model of long-range regulation by non-genic SNPs [19], and potentially add to the evolving concept of transcriptional regulation by intergenic regions and perturbations introduced by their polymorphisms [26].

### 4.3. Adipocytokines, Cancer and OSAHS

Adipocytokines signaling and cancer, as they arise in our analyses and are corroborated by others [19,20,21,26] should be considered as cross-talking pathophysiological substrates, further enhanced by OSAHS. The relationship between OSAHS and adipocytokines may arise from a common genetic substrate, as the recently discovered interplay between OSAHS and hypertriglyceridaemia has shown [27].

Adipocytokine cascades are well established clinical correlates with cancer [26], with obesity-related proteins in general being associated with an increased risk of breast cancer in females [28]. In this setting, adipocytokine signaling and hypoxia combined in the setting of a metabolically active adipose tissue creates favorable conditions for tumorigenesis, including sustained proinflammatory signaling and remodeling of the extracellular matrix [29].

### 4.4. Limitations and Strengths

Our findings should be interpreted within the conceptual framework of this study’s limitations and strengths. A major limitation of our study is that the clinical cohort did not achieve sufficient sample size, reflected by the invariance of rs116133558 and rs75108997. As such, we were not able to sufficiently detect associations between genotype frequency and OSAHS parameters at the clinical level. Considering that recruitment was greatly affected due to the pandemic, this obstacle could not be overcome within the given timeframe of our study. Another important caveat regarding the clinical cohort is that as a nested study, the reference population is specific and thus any finding we could have drawn would be of reduced generalization. Furthermore, while we reconstructed a regulatory gene network from two intergenic SNPs, we cannot confirm our findings in a prospective cohort. We overcome this limitation by comparing our findings with two of the most comprehensive studies in OSAHS and we managed to validate our major findings both on the genomic and pathway level.

## 5. Conclusions

This is the first study to uncover a regulatory network in OSAHS using transcriptional targets of intergenic SNPs and map their contributions in the pathophysiology of the syndrome on the interplay between adipocytokine signaling and cancer-related transcriptional dysregulation. Further studies are needed to expand this concept in other intergenic SNPs and outline the non-genic networks governing long-range transcriptional regulation in OSAHS.

## Figures and Tables

**Table 1 diagnostics-11-01753-t001:** Demographic, and clinical characteristics.

	OSAHS (*n* = 102)	Controls (*n* = 50)
Age	56.99 ± 13.1	53.5 ± 9.6
Male/Female	74 (72.5%)/28 (27.5%)	19 (38%)/31 (62%)
BMI	32.83 ± 7.16	26.56 ± 3.6
AHI	42.57 ± 24.38	38.30 ± 25.43
Smokers	41 (40.2%)	10 (20%)
Diabetes	11 (10.8%)	0 (0%)
Hyperlipidemia	52 (51%)	9 (18%)
Hypertension	61 (59.8%)	9 (18%)
Coronary heart disease	17 (16.7%)	1 (2%)

BMI: Body Mass index.

**Table 2 diagnostics-11-01753-t002:** Biochemical and PSG parameters.

	Controls (*n* = 50)	OSAHS (*n* = 102)	*p*-Value
Glucose	93.08 ± 10.41	10.42 ± 18.82	0.531
Cholesterol	168.56 ± 61.80	171.84 ± 35.03	0.728
Triglyceride	140.0000 ± 85.41	142.4608 ± 54.74	0.830
HDL	51.5000 ± 10.22	46.2529 ± 8.44	0.001
LDL	119.1240 ± 33.86	113.7039 ± 35.40	0.370
CRP	0.26 ± 0.24	0.87 ± 0.92	<0.001
AHI	3.48 ± 0.63	38.30 ± 25.43	<0.001
ODI	2.80 ± 1.17	40.89 ± 30.71	<0.001
TST	7.44 ± 1.50	7.21 ± 1.15	0.334
TST90%	0.61 ± 0.76	29.67 ± 44.49	<0.001
AI	0.67 ± 0.53	15.78 ± 20.26	<0.001

HDL: high-density lipoprotein; LDL: low-density lipoprotein; CRP: C-reactive protein; TST: total sleep time in hours; TST90%: percentage of total sleep time in hours under 90% SaO_2_; AI: Arousal Index.

**Table 3 diagnostics-11-01753-t003:** Binary logistic regression model (BLRM) with G/A as the reference group for the rs999944.

B	S.E.	Wald	df	Sig.	Exp(B)	95% C.I. for EXP(B)		
						Lower	Upper	
Sex	−1.019	0.521	3.826	1	0.050	0.361	0.130	1.002
Age	0.027	0.021	1.615	1	0.204	1.027	0.986	1.070
BMI	0.001	0.042	0.001	1	0.978	1.001	0.922	1.087
OSAHS Group			3.091	3	0.378			
Control	0.310	0.710	0.191	1	0.662	1.364	0.339	5.485
Mild to Moderate OSAHS	1.011	0.974	1.077	1	0.299	2.747	0.407	18.526
Severe OSAHS	1.261	0.784	2.588	1	0.108	3.530	0.759	16.413
Glucose	0.040	0.017	5.352	1	0.021	1.041	1.006	1.077
Cholesterol	−0.004	0.007	0.288	1	0.592	0.996	0.983	1.010
Triglyceride	0.002	0.004	0.193	1	0.660	1.002	0.994	1.010
HDL	0.008	0.029	0.074	1	0.785	1.008	0.952	1.068
LDL	0.007	0.009	0.654	1	0.419	1.007	0.990	1.024
CRP	−0.196	0.351	0.311	1	0.577	0.822	0.413	1.636
Constant	−3.280	2.752	1.421	1	0.233	0.038		

Variable(s) entered on step 1: gender, age, BMI, OSAHSGroup, glucose, cholesterol, triglycerides, HDL, LDL, CRP.

**Table 4 diagnostics-11-01753-t004:** GeneCARDS Results For Pseudogene Gene Targets and Transcription-Factor Binding Sites.

SNP (RSID)	Pseudogene Accession	GH Type	BLAST % Identity	Gene Targets	TFBS	GH Score
rs116133558	AC114402.1	enhancer	100%	ENSG00000227417, LOC100420338, SNRPE, LAX1, ZBED6, ZC3H11A, ENSG00000223505	CEBPB, FOS, JUND, RAD21	0.6
rs999944	AC007880.2	promoter, enhancer	100%	lnc-SLC1A4-5, VPS54, SERTAD2, RN7SL211P, LINC02576, SLC1A4, RAB1A, LINC01800, ACTR2, AFTPH, ENSG00000199964	256 TFs	1.8

SNP: single nucleotide polymorphism, TF: transcription factor, tfbs: transcription-factor binding site.

**Table 5 diagnostics-11-01753-t005:** The 10 first pathways associated with I_A_—WikiPathways.

Index	Pathway	*p*-Value	Adj.*p*-Value
1	prion disease pathway WP3995	3.815 × 10^−18^	1.026 × 10^−15^
2	androgen receptor signaling pathway WP138	4.897 × 10^−15^	6.587 × 10^−13^
3	adipogenesis WP236	2.142 × 10^−13^	1.887 × 10^−11^
4	TGF-beta signaling pathway WP366	2.806 × 10^−13^	1.887 × 10^−11^
5	sudden infant death syndrome (SIDS) susceptibility pathways WP706	6.118 × 10^−13^	3.292 × 10^−11^
6	The effect of progerin on the involved genes in Hutchinson–Gilford progeria syndrome WP4320	1.628 × 10^−12^	7.299 × 10^−11^
7	circadian rhythm related genes WP3594	5.297 × 10^−12^	2.036 × 10^−10^
8	integrated breast Cancer pathway WP1984	3.026 × 10^−11^	1.017 × 10^−9^
9	nuclear receptors WP170	6.664 × 10^−11^	1.992 × 10^−9^
10	hematopoietic stem cell gene regulation by GABP alpha/beta complex WP3657	1.224 × 10^−10^	3.292 × 10^−9^

**Table 6 diagnostics-11-01753-t006:** The 10 first pathways associated with I_A_—KEGG.

Index	Pathway	*p*-Value	Adj.*p*-Value
1	transcriptional misregulation in cancer	6.270 × 10^−19^	8.652 × 10^−17^
2	pathways in cancer	1.167 × 10^−9^	8.053 × 10^−8^
3	osteoclast differentiation	2.292 × 10^−9^	1.054 × 10^−7^
4	human T-cell leukemia virus 1 infection	9.866 × 10^−9^	3.404 × 10^−7^
5	viral carcinogenesis	1.898 × 10^−8^	5.237 × 10^−7^
6	acutemyeloid leukemia	2.789 × 10^−7^	5.499 × 10^−6^
7	Th17 cell differentiation	2.530 × 10^−7^	5.499 × 10^−6^
8	thyroidcancer	6.078 × 10^−7^	1.048 × 10^−5^
9	hepatocellular carcinoma	3.648 × 10^−6^	5.594 × 10^−5^
10	thyroid hormone signaling pathway	4.501 × 10^−6^	5.647 × 10^−5^

## Data Availability

The data that support the findings of this study are available on request from the corresponding author. The data are not publicly available due to containing information that could compromise the privacy of research participants.

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
