# Peer review of "Intergenic SNPs in Obstructive Sleep Apnea Syndrome: Revealing Metabolic, Oxidative Stress and Immune-Related Pathways"

_diagnostics, 2021, doi:10.3390/diagnostics11101753_

Round 1

Reviewer 1 Report

Thank you very much for the current version. In my opinion, it may be published in the present form.

Author Response

Thank you for your comment, and the appraisal of our work.

Reviewer 2 Report

I would like to thank the authors for observing my reviews and carefully reviewing the manuscript.

MAJOR POINTS

  • The logic behind the selection of the four candidate SNPs is much clearly stated in the introduction section of the revised manuscript. 

  • It is absolutely clear that that the attempt was to "replicate" the results of the study by Cade et al, by assessing in a new population  the four SNPs that were reported as associated with some components of OSAHS in the original publication. Even when considered as a 'standalone approach', as suggested in the first rebuttal point,  the result of such assessment was that there is no statistically significant association between the genotype frequency  rs999944 SNP  and OSAHS compared with controls Hence further characterization of this SNP is scientifically irrelevant for this study. Even when in silico SNP characterization can be pursued regardless the clinical cohort (i.e.  second rebuttal point), in this case it would represent an interpretation of the function for  the SNPs by Cade et al. ; but it will not add any relevance to the study reported in this manuscript just because there is no association.

  • Long genomic regions are certainly used to map SNPs and detect recombination. On the other hand, for "SNP discovery", it is acceptable to screen long regions. However, if the goal is to infer the function of a SNP by associating with some structural, functional or regulatory element the exact location of the SNP must be evaluated, unless of course it is known distal regulator (e.g. an enhancer). In this study in particular, the SNPs are well mapped  with no need for "discovery", but rather a variant interpretation.

MINOR POINTS

  • Typos and/or grammatic mistakes still present in the manuscript.
  • Still some commas are used to separate decimals in table 1

Author Response

  • The logic behind the selection of the four candidate SNPs is much clearly stated in the introduction section of the revised manuscript. 

Thank you for your comment.

  • It is absolutely clear that that the attempt was to "replicate" the results of the study by Cade et al, by assessing in a new population the four SNPs that were reported as associated with some components of OSAHS in the original publication. Even when considered as a 'standalone approach', as suggested in the first rebuttal point,  the result of such assessment was that there is no statistically significant association between the genotype frequency  rs999944 SNP  and OSAHS compared with controls Hence further characterization of this SNP is scientifically irrelevant for this study. Even when in silico SNP characterization can be pursued regardless the clinical cohort (i.e.  second rebuttal point), in this case it would represent an interpretation of the function for  the SNPs by Cade et al. ; but it will not add any relevance to the study reported in this manuscript just because there is no association.

Thank you for your comment. While we agree, as we have reported, that our study did not reveal associations in this clinical cohort, an in-silico investigation using data that are openly accessible (and hence, independent) presents added value to the potential biological relevance of this SNP. Considering that this has not been previously performed via the approaches we utilize, it represents novel information.

Furthermore, understanding an SNPs’ potential function also helps characterize the factors and mechanisms that would be absent from our own cohort. Considering OSAHS’ genetic and phenotypic heterogeneity, the in silico investigation of an absent factor still provides context. Our rationale in pursuing this characterization that less information on an SNP, especially when this applies to the literature as well, will not improve or present added value to an investigation on said SNP in a potentially underpowered cohort.

  • Long genomic regions are certainly used to map SNPs and detect recombination. On the other hand, for “SNP discovery”, it is acceptable to screen long regions. However, if the goal is to infer the function of a SNP by associating with some structural, functional or regulatory element the exact location of the SNP must be evaluated, unless of course it is known distal regulator (e.g. an enhancer). In this study in particular, the SNPs are well mapped  with no need for “discovery”, but rather a variant interpretation.

Thank you for your comment. We agree with the reviewer that is generally the case for SNPs that alter gene functions, and thus, their specific location should be evaluated. For intergenic SNPs however, their specific location does not provide extra information on its own; considering that long distance regulation and other relationships both upstream and downstream have been shown to present important targets that reveal their biological functions. For example, Schierding and colleagues1 provide such a case in point for intergenic SNPs that could affect gene expression up to 85kb away from the target gene. We have rephrased discovery into investigation, to better characterize our aim and the application of methods, and added the reference presented here to support and better present our rationale.

  1. Schierding W, Antony J, Cutfield WS, Horsfield JA, O'Sullivan JM. Intergenic GWAS SNPs are key components of the spatial and regulatory network for human growth. Hum Mol Genet. 2016;25(15):3372-3382. doi:10.1093/hmg/ddw165

Reviewer 3 Report

I have read the article by Raptis et al. with great interest. The authors analysed four SNPs in OSA which may be related to metabolic disease and cancer.

Comments:

  • Throughout the manuscript the authors use OSAHS, whilst I feel they really mean OSA (without syndrome). I think using OSA is righter. Please amend it accordingly.
  • Could you please explain in detail the criteria of scoring hypopnoeas?
  • Please add power calculations.
  • Control group. You mentioned that they were healthy in the Methods, yet in the results some of them suffered from comorbidities. Please, clarify.
  • Table 1. The mean and SD values for age should be written with 1 decimal maximum.
  • Table 1. Please, provide p values for comparisons.
  • Table 1. I see that lipids, CRP and glucose were measured. Please, provide these results in Table 1 and compare them between the two groups.
  • Table 1. PSG was performed. Yet, I do not see any data and the most important of them (i.e. TST, AHI, ODI, TST90%, arousal index) should be compared between the groups.
  • 4.1. The authors discuss the genetic relationship between adipokine production and OSA. I believe the authors should consider citing https://pubmed.ncbi.nlm.nih.gov/31908118/ which revealed that OSA and hypertriglyceridemia are genetically linked.
  • Line 17. “Were performed BLAST”. I guess you meant “we” rather than “were”

Author Response

  • Throughout the manuscript the authors use OSAHS, whilst I feel they really mean OSA (without syndrome). I think using OSA is righter. Please amend it accordingly.

Thank you for the comment. As the patients fulfilled the criteria for the syndrome, we amended OSA to OSAHS as per your suggestion.

  • Could you please explain in detail the criteria of scoring hypopnoeas?

Thank you for your comment. The scoring criteria for hypopneas have been explained in the methods, after the definition of apneas.

  • Please add power calculations.

Thank you for your suggestion. We have added post-hoc power analyses to further indicate that our clinical cohort was under-powered.

  • Control group. You mentioned that they were healthy in the Methods, yet in the results some of them suffered from comorbidities. Please, clarify.

Thank you for your comment. We have renamed healthy controls to “non-OSAS” controls, in order to reflect incident comorbidities.

  • Table 1. The mean and SD values for age should be written with 1 decimal maximum.

Thank you for your suggestion. This has been amended.

  • Table 1. Please, provide p values for comparisons.

Thank you for your suggestion. This has been amended.

  • Table 1. I see that lipids, CRP and glucose were measured. Please, provide these results in Table 1 and compare them between the two groups.

Thank you for your comment. We have added these comparisons in table 2.

  • Table 1. PSG was performed. Yet, I do not see any data and the most important of them (i.e. TST, AHI, ODI, TST90%, arousal index) should be compared between the groups.

Thank you for your comment. We have added these comparisons in table 2.

  • 1. The authors discuss the genetic relationship between adipokine production and OSA. I believe the authors should consider citing https://pubmed.ncbi.nlm.nih.gov/31908118/ which revealed that OSA and hypertriglyceridemia are genetically linked.

Thank you for your suggestion. The study was highly relevant, and we have added it in the adipokine – inflammation segment in the discussion.

  • Line 17. “Were performed BLAST”. I guess you meant “we” rather than “were”

Thank you for your comment. We have restructured the entire sentence for clarity and brevity.

Round 2

Reviewer 2 Report

Thanks to the authors for considering my points.

I concur with the authors on the favorable arguments for in silico characterization of the SNPs. However, the major point is whether further investigation of a SNP which is not associated with the trait in the population under study is scientifically relevant.

I noted that the authors expanded the work by investigating other publicly available datasets. The justification of why this particular SNP is chose is still unclear. 

In any case, if the authors choose to pursue this research line, I consider that results must be presented in a completely different design (i.e. meta-analysis) and not an ad hoc addition to the present dataset.

Reviewer 3 Report

I am happy that the authors implemented the requested changes.

Table 2. I think the values in the OSA and control groups should be the other way around.